# Protecting young children from future pandemics: getting the basics right

Chunling Lu  ,[1,2] Jere R Behrman,[3] Linda M Richter[4]

The United Nations Convention on the Rights of the Child has defined four basic rights of children: to survival, to protection, to development and to participation.[1] Protecting young children from the direct and secondary impacts of the COVID-19 and future pandemics and ensuring their survival and healthy development requires progress in both technology and household socioeconomic infrastructure.

Since the outbreak of COVID-19, a number of global initiatives have been established, primarily aimed at coordinating global-level, regional-level and country-level efforts to develop, produce, distribute and improving uptake of biomedical technologies to fight the pandemic.[2] The goal is to stop the pandemic and rapidly recover economic and societal activities by accelerating the development of vaccines, therapeutics and diagnostics and supporting, especially low-income and middle-income countries (LMICs), in translating newly developed tools into national responses to the pandemic. Efforts and achievements of global partnerships for combating COVID-19 are applauded as the 'world's most comprehensive end-to-end solution'.[3] The models used for coordinating actions between unprecedentedly large numbers of stakeholders offer invaluable lessons and experiences for preparation for future pandemics.

Despite these exciting achievements, it is important to note that global activities have focused mainly on enhancing biomedical capacity for disease control; little attention has been given to improving social conditions by helping millions of households to be better prepared for future pandemics. Social conditions are very important for pandemics such as

COVID-19, as for most health challenges; in some cases more important than biomedical capacities for disease control.

We argue that developing and distributing new medical technologies is necessary, but not sufficient for conquering future pandemics. As recommended by the WHO,[4] for viral infections caused by respiratory droplets, physical contacts and contaminated food or water, ensuring households have basic sanitation, hygiene facilities, clean water, ability to quarantine and access to information is crucial for effective prevention and safe care at home. Strengthening household living conditions and their socioeconomic infrastructure should be included in global preparation as an essential component, especially given the lag between discovery and mapping of a disease, and invention or adaptation of tools for controlling it.

Developing new technologies requires knowledge about the virus and long-time scientific research on robust methods of delivery. The success of developing the messenger RNA COVID-19 vaccine within a short time built on decades of rigorous scientific investigations; nevertheless, the lag from virus discovery to the first vaccine delivery was a year.[5] Before those tools became available, we were reliant on prevention methods, such as home quarantine and hygiene practices, to limit viral spread. Additionally, even after vaccines or treatments are developed, ensuring equal access to these tools for populations in need, especially those in LMICs, has proven to be very difficult due to technical, operational, financial and informational challenges. Even in

countries with high rates of vaccination, home quarantine, handwashing, physical distancing and masks are still required to curb transmission. Empowering households with the means for preventing the spread of a pandemic remains pivotal to the success of reducing prevalence and mortality.

Empirical evidence about young children living in households with preparedness for COVID-19 in LMICs is worrying. In our investigation of 56 LMICs with available nationally representative data,[6] on average <20% of young children lived in households that meet five preparedness conditions for preventing COVID-19 or communicable diseases (box 1). Children in low-income countries or sub-Saharan Africa were the most disadvantaged—only 4.4% or 4.6%, respectively, lived in prepared households. In most countries, significant residence-disparities and/or wealth-disparities in household preparedness were observed, favouring children living in urban areas or in the richest households. For example, in Paraguay, the gap in young children living in prepared households between the poorest (10.7%) and the richest (90.2%) was close to 80 percentage points. Lacking access to basic sanitation and/or hygiene facilities are the two main contributors to poor household preparation in the 56 countries. Evidence in high-income countries showed that minorities living in overcrowded households with poor hygiene had higher transmission risk of COVID-19 than their counterparts.[7] These findings suggest that home care or confinement is currently beyond the capacity of many low-income households, potentially compounding a range of health risks to young children when they were locked down at home by government regulations.

Ensuring everyone (including children) have access to clean water, safe basic sanitation and hygiene by 2030 is one of the Sustainable Development Goals (SDG 6),[8]

[1]Division of Global Health Equity, Brigham and Women's Hospital, Boston, Massachusetts, USA
[2]Department of Global Health and Social Medicine, Harvard Medical School, Boston, Massachusetts, USA
[3]Department of Economics, University of Pennsylvania, Philadelphia, Pennsylvania, USA
[4]DSI-NRF Centre of Excellence in Human Development, University of the Witwatersrand Johannesburg, Johannesburg, South Africa

**Correspondence to** Dr Chunling Lu; Chunling_Lu@hms.harvard.edu

---

**Box 1　Household preparedness for the COVID-19 and future pandemics[5]**

1. Access to basic hygienic facilities: measured by availability of handwashing facilities with water and soap.
2. Access to basic sanitation facilities: measured by access to sanitation not shared with other households and with hygienic separation of excreta from human contact (eg, flush to piped sewer system/septic tank/pit latrine, ventilated improved pit, pit latrine with slab).
3. Adequate quarantine conditions: measured by three persons or less per sleeping room.
4. Being connected to outside support: measured by household ownership of at least one phone (landline or mobile).
5. Access to up-to-date information about the pandemics: measured by mass-media exposure among caregivers.

though progress in achieving SDG 6 in low-income settings has been very slow. In sub-Saharan Africa, the average population coverage of basic sanitation increased from 23% in 2000 to 30% in 2017, and only 25% lived in households with handwashing facilities in 2017.[8] As extreme weather becomes more frequent due to the climate change, delivering water, sanitation and hygiene (WASH) services is more challenging in low-income countries. The importance of these services in protecting households from infections and the slow progress and increasing challenges in coverage requires immediate attention and action from governments, global health and financial organisations, donors and other stakeholders.

In many LMICs, current investments in WASH are far from sufficient for achieving SDG 6. Low-income countries have limited domestic tax revenue for implementing their plans on WASH due to their low economic development as well as revenue leaks, including global tax abuse facilitated by high-income countries in some respects. It has been estimated that increasing governmental revenues equivalent to global tab tax abuse could be associated with 36 million more people having access to basic sanitation and 18 million more to basic drinking water in LMICs.[9] However, aid for WASH in LMICs has experienced a fall since the SDGs era (by 5.6% between 2017 and 2020).[10] To prepare households in LMICs for future pandemics, high-income countries should increase their aid for WASH, providing compensation to LMICs for climate damages caused by global warming, and help LMICs achieve their targets on SDG 6.

To summarise, ensuring children's basic rights during the pandemics requires improvement of their living environment. Getting low-income households prepared for future pandemics should be part of the global preparation strategies and deserves more political commitments and financial investments.

**Contributors** CL and LMR conceptualised the paper. DL drafted the paper. All authors revised, finalised and approved the paper.

**Funding** UKRI GCRF Collective Fund Award (Grant Ref: ES/T003936/1) to the University of Oxford, UKRI ECRS GCRF, Harnessing the power of global data to support young children's learning and development: Analyses, dissemination and implementation.

**Competing interests** None.

**Patient consent for publication** Not applicable.

**Ethics approval** Not applicable.

**Provenance and peer review** Not commissioned; externally peer reviewed.

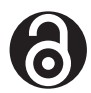

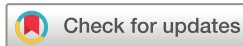

**To cite** Lu C, Behrman JR, Richter LM. *BMJ Paediatrics Open* 2023;7:1–2.

*BMJ Paediatrics Open* 2023;7:1–2.

**ORCID iD**
Chunling Lu http://orcid.org/0000-0002-4780-9451

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
