## [Reviewer comments · BMJ Paediatrics Open]

ARTICLE DETAILS

TITLE (PROVISIONAL)	Protecting young children from future pandemics: Getting the basics right
AUTHORS	Lu, Chunling Behrman, Jere Richter, Linda

VERSION 1 - REVIEW

REVIEWER	Reviewer Name: Dr. Nick Spencer Institution and Country: University of Warwick, United Kingdom of Great Britain and Northern Ireland
REVIEW RETURNED	16-Apr-2023

GENERAL COMMENTS	This Viewpoint addresses the need to combine medical technological approaches, such as vaccines, with measures to improve the social determinants of health, such as shelter, sanitation, clean water etc. The technical progress and collaboration shown in the development of COVID-19 vaccines in response to the pandemic was impressive but, as the authors of this Viewpoint argue, technical responses were necessary but not sufficient for dealing with the COVID-19 pandemic and will not be sufficient for future pandemics. The SARS-COV-2 virus spread rapidly through populations and its transmission was assisted by overcrowded living conditions with poor ventilation and inadequate hygiene facilities. Poorer populations within and between countries were most affected. The authors cite evidence for lack of household preparedness for COVID-19 in low- and middle-income countries (LMICs) and similarly poor hygiene and overcrowding are thought to partially explain increased infection rates in disadvantaged areas of the UK. This article is a timely contribution to the literature on the lessons of the COVID-19 pandemic for future pandemics as well as illustrating the continued deprivation of the basic needs for healthy growth and development of many of the world's children. I have one specific suggestion for strengthening the argument for action from high-income country governments to improve living conditions for children in LMICs and a few minor edits: 1. Global tax abuse (avoidance and evasion), enabled by rich nations, contributes significantly to reducing the budgets of LMICs preventing them from making necessary improvements to the basic determinants of health. We have estimated that the increase in government revenue equivalent to global tax abuses is associated with 36 million people having access to basic sanitation and 18 million having access to basic drinking water allowing countries to achieve their Sustainable Development Goals [1]. I know the authors are limited to 5 references; however, I think tax abuse is a major factor limiting improvement in basic determinants in LMICs and would greatly strengthen the call for action in the article.
---

	2. Minor edits: p.2 line28 – ‘s’ missing from households p.2 line 34- suggest start new paragraph at ‘Developing’ p.3 Ref 1 - incomplete [1] O’Hare BAM, Lopez MJ, Mazimbe B, Murray S, Spencer N, Torrie C, et al. (2022) Tax abuse—The potential for the Sustainable Development Goals. PLOS Glob Public Health 2(2):e0000119. https://doi.org/10.1371/journal.pgph.0000119
--	--

REVIEWER	Reviewer Name: Dr. Simon Lenton Institution and Country: United Kingdom of Great Britain and Northern Ireland
REVIEW RETURNED	21-Apr-2023

GENERAL COMMENTS	An excellent article. While “children” appear in the title there is relatively little reference to children and the article itself. It might be improved by reference to human rights/United Nations Convention on the Rights of the Child which can be distilled down to 4 principles, health promotion, protection, provision and participation each of which is relevant. The could be more explicit reference to the inequity in outcomes for COVID in high-income countries. It might be interesting to compare and contrast the costs of providing basic sanitation and COVID costs (I think both are approximately \$100b). A reference to the impact of climate change and reparations for low income countries to improve standards of living might also be included.
--

REVIEWER	Reviewer Name: Dr. Luis Rajmil Institution and Country: Spain
REVIEW RETURNED	25-Apr-2023

GENERAL COMMENTS	The opinion article reflects an important factor. Perhaps the following aspects could be taken into account in order to improve the presentation:  - According to the Authors, developing and distributing new medical technologies is necessary, but not sufficient for conquering the future pandemics. - Habitability conditions are also fundamental, it is true, but they are also a reflection of social inequalities and social gradients, which exist in all countries, especially, as the authors explain, in developing countries. - It would be interesting if the model is extended to all determinants of social inequalities, that they have also had an influence on the results of the pandemic such as unemployment, level of family education, level of family income, etc.
---

VERSION 1 – AUTHOR RESPONSE

Dear editor and reviewers,

We sincerely thank you and the reviewers for your valuable comments. We are submitting a revised clean version and providing point-by-point replies below to your requests. We have highlighted the changes in the submitted file “Main Document – marked copy”. We look forwards to your decision and are happy to consider further revisions if suggested.

(1) The five preparedness conditions listed should be placed in a Box or Table to emphasise their importance.

We have added a panel, as below, to list the five conditions of household preparedness for the COVID-19 and future pandemics.

Panel 1 Household preparedness for the COVID-19 and future pandemics⁴

Access to basic hygienic facilities: measured by availability of handwashing facilities with water and soap

Access to basic sanitation facilities: measured by access to sanitation not shared with other households and with hygienic separation of excreta from human contact (e.g., flush to piped sewer system/septic tank/pit latrine, ventilated improved pit, pit latrine with slab, etc.)

Adequate quarantine conditions: measured by three persons or less per sleeping room

Being connected to outside support: measured by household ownership of at least one phone (landline or mobile)

Access to up-to-date information about the pandemics: measured by mass-media exposure among caregivers

(2) Mention the Sustainable Development Goal 6 in relation to clean water and sanitation

We have included the SDGs 6 in the text (as below) and added the related citation in the “References”.

“Ensuring everyone (including children) having access to clean water, save basic sanitation, and hygiene by 2030 is one of the Sustainable Development Goals (SDG 6),⁷ though progress in achieving SDG 6 in low-income settings has been very slow.”

Comments from the reviewers:

Reviewer #1 Dr. Nick Spencer

This Viewpoint addresses the need to combine medical technological approaches, such as vaccines, with measures to improve the social determinants of health, such as shelter, sanitation, clean water etc. The technical progress and collaboration shown in the development of COVID-19 vaccines in response to the pandemic was impressive but, as the authors of this

Viewpoint argue, technical responses were necessary but not sufficient for dealing with the COVID-19 pandemic and will not be sufficient for future pandemics.

The SARS-COV-2 virus spread rapidly through populations and its transmission was assisted by overcrowded living conditions with poor ventilation and inadequate hygiene facilities. Poorer

populations within and between countries were most affected. The authors cite evidence for lack of household preparedness for COVID-19 in low- and middle-income countries (LMICs) and similarly poor hygiene and overcrowding are thought to partially explain increased infection rates in disadvantaged areas of the UK.

This article is a timely contribution to the literature on the lessons of the COVID-19 pandemic for future pandemics as well as illustrating the continued deprivation of the basic needs for healthy growth and development of many of the world's children. I have one specific suggestion for strengthening the argument for action from high-income country governments to improve living conditions for children in LMICs and a few minor edits:

(1) Global tax abuse (avoidance and evasion), enabled by rich nations, contributes significantly to reducing the budgets of LMICs preventing them from making necessary improvements to the basic determinants of health. We have estimated that the increase in government revenue equivalent to global tax abuses is associated with 36 million people having access to basic sanitation and 18 million having access to basic drinking water allowing countries to achieve their Sustainable Development Goals [1]. I know the authors are limited to 5 references; however, I think tax abuse is a major factor limiting improvement in basic determinants in LMICs and would greatly strengthen the call for action in the article.

[1] O'Hare BAM, Lopez MJ, Mazimbe B, Murray S, Spencer N, Torrie C, et al. (2022) Tax abuse—The potential for the Sustainable Development Goals. PLOS Glob Public Health 2(2):e0000119. <https://doi.org/10.1371/journal.pgph.0000119>

We appreciate Dr. Spencer's comments and thank him for providing us with the reference. In response to the comments from Dr. Spencer and other reviewers on investing household infrastructural conditions, we added the following discussions on financing SDG 6.

"In many LMICs, current investments in water, sanitation, and hygiene (WASH) are far from sufficient for achieving SDG 6. Low-income countries have limited domestic tax revenue for implementing their plans on WASH due to their low economic development as well as revenue leaks, including global tax abuse facilitated in some respects by high-income countries. It has been estimated that increasing governmental revenues equivalent to global tax abuse could be associated with 36 million more people having access to basic sanitation and 18 million more to basic drinking water in LMICs.⁸ However, aid for WASH in LMICs has experienced a fall since the SDGs era (by 5.6% between 2017 and 2020).⁹ To prepare households in LMICs for future pandemics, high-income countries should increase their aid for WASH, providing compensation to LMICs for climate damages by global warming, and help LMICs achieve their targets on SDG 6."

(2). Minor edits:

p.2 line28 – 's' missing from households

p.2 line 34- suggest start new paragraph at 'Developing

p.3 Ref 1 – incomplete

Thank you! These edits have been done.

Reviewer # 2 Dr. Simon Lenton

An excellent article.

(1) While "children" appear in the title there is relatively little reference to children and the article itself. It might be improved by reference to human rights/United Nations Convention on the Rights of

the Child which can be distilled down to 4 principles, health promotion, protection, provision and participation each of which is relevant.

We appreciate comment and has added the following discussions about young children in the text.

“The United Nations Convention on the rights of the child has defined four basic rights of children: to survival, to protection, to development, and to participation.¹ Protecting young children from the direct and secondary impacts of the COVID-19 and future pandemics and ensuring their survival and healthy development requires progress in both technology and household socioeconomic infrastructure.”

(2) There could be more explicit reference to the inequity in outcomes for COVID in high-income countries.

We have added the following discussions in the text.

“Evidence in high-income countries showed that minorities living in overcrowded households with poor hygiene had higher transmission risk of COVID-19 than their counterparts.⁶”

(3) It might be interesting to compare and contrast the costs of providing basic sanitation and COVID costs (I think both are approximately \$100b).

We added the following sentences to discuss the importance of investment in WASH.

“In many LMICs, current investments in water, sanitation, and hygiene (WASH) are far from sufficient for achieving SDG 6. Low-income countries have limited domestic tax revenue for implementing their plans on WASH due to their low economic development as well as revenue leaks, including global tax abuse facilitated in some respects by high-income countries. It has been estimated that increasing governmental revenues equivalent to global tax abuse could be associated with 36 million more people having access to basic sanitation and 18 million more to basic drinking water in LMICs.⁸ However, aid for WASH in LMICs has experienced a fall since the SDGs era (by 5.6% between 2017 and 2020).⁹ To prepare households in LMICs for future pandemics, high-income countries should increase their aid for WASH, providing compensation to LMICs for climate damages by global warming, and help LMICs achieve their targets on SDG 6.”

(4) A reference to the impact of climate change and reparations for low income countries to improve standards of living might also be included.

“As extreme weather becomes more frequent due to climate change, delivering WASH services is more challenging in low-income countries.”

“To prepare households in LMICs for future pandemics, high-income countries should increase their aid for WASH, providing compensation to LMICs for climate damages caused by global warming, and help them achieve their targets on SDG 6.”

Reviewer 3. Dr. Luis Rajmil

The opinion article reflects an important factor. Perhaps the following aspects could be taken into account in order to improve the presentation:

- According to the Authors, developing and distributing new medical technologies is necessary, but not sufficient for conquering the future pandemics.

- Habitability conditions are also fundamental, it is true, but they are also a reflection of social inequalities and social gradients, which exist in all countries, especially, as the authors explain, in developing countries.

- It would be interesting if the model is extended to all determinants of social inequalities, that they have also had an influence on the results of the pandemic such as unemployment, level of family education, level of family income, etc.

We appreciate the reviewer's comments and have added the related findings about socioeconomic inequalities in household preparedness.

"In most countries, significant residence- and/or wealth-disparities in household preparedness were observed, favoring children living in urban areas or in the richest households. For example, in Paraguay, the gap in young children living in prepared households between the poorest (10.7%) and the richest (90.2%) was close to 80 percentage points."

VERSION 2 – REVIEW

REVIEWER	Reviewer Name: Dr. Nick Spencer Institution and Country: University of Warwick, United Kingdom of Great Britain and Northern Ireland
REVIEW RETURNED	08-May-2023

GENERAL COMMENTS	The authors have comprehensively and appropriately responded to all the reviewers' comments. The paper is now acceptable for publication in the journal.
--

REVIEWER	Reviewer Name: Dr. Simon Lenton Institution and Country: United Kingdom of Great Britain and Northern Ireland
REVIEW RETURNED	08-May-2023

GENERAL COMMENTS	all good
----------

VERSION 2 – AUTHOR RESPONSE